# HYPERBOLIC MUSIC REPRESENTATIONS

## ABSTRACT

Music is inherently hierarchical due to keys and variations of note sequences. These dependencies need to be captured by the metric of choice to learn an appropriate representation space. Although Euclidean geometry is frequently used to embed music, it is clearly unable to capture the hierarchical structures. In this paper, we propose to learn hyperbolic representation spaces for music using Variational Autoencoders with a Poincaré ball as a natural alternative to Euclidean geometry. The resulting latent space is interpretable, reflects keys and musical richness, and allows for meaningful interpolations due to a novel generalization of Spherical Linear Interpolation to Riemannian manifolds. Empirically, we compare our contribution to standard Euclidean representations and observe that the latter fall short in terms of interpretation and reconstruction.

## 1 INTRODUCTION

In the wake of foundation models, the generation of music has recently become a topic of interest to a constantly growing community. Over the last few years, several models have been proposed to generate high-quality audio samples that create desired pieces of music by conditioning the generation process on annotations like genre or artist (Dhariwal et al., 2020), language prompts (Agostinelli et al., 2023; Bhandari et al., 2025), or melodic inputs (Copet et al., 2023).

An alternative to controllable music generation is offered by variational autoencoders (VAEs) which represent music in compact latent spaces (Roberts et al., 2018; Brunner et al., 2018; Caillon & Esling, 2021). New compositions can be generated by modifying latent representations, allowing for a rather direct form of control. However, VAEs operate on Euclidean geometry and rely on the assumption that the data manifold in input space is intrinsically flat.

When this assumption is violated, representations in the latent space will necessarily be distorted. This follows from Gauss's Theorema Egregium, stating that the Gaussian curvature of a surface is invariant under isometric mappings (Gauss, 1828). Consequently, a mapping onto another manifold with a different curvature cannot be an isometry.

In this paper, we provide evidence that music is naturally hierarchical. Intuitively, hierarchical relations between pieces of music emerge because the combinatorial number of possible ways to extend musical sequences grows with the number of the involved unique notes: A melody, consisting of a few notes, can be extended in multiple ways by adding only one additional note, whereby the original melody would be contained as a subset of every possible extension. This observation renders Euclidean embeddings of music in general inappropriate and instead gives rise to hyperbolic geometry that suggests itself for embedding hierarchical structures (Nickel & Kiela, 2017).

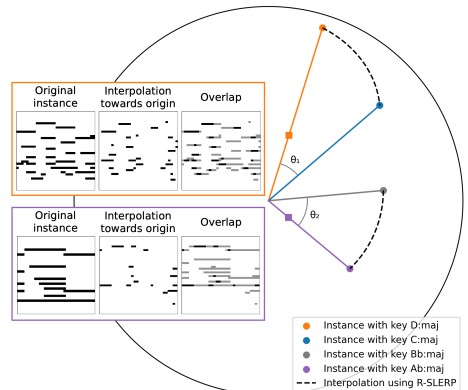

Figure 1: Two pairs of melodies encoded in the Poincaré ball, demonstrating the musical richness depending on the distance to origin while the keys vary consistently with the angles $\theta_1$ and $\theta_2$. The dotted lines show the newly proposed interpolation scheme R-SLERP.

To embed music in hyperbolic space, we develop a music VAE with a Poincaré ball as latent space. Figure 1 illustrates how the described hierarchy is organized on the ball. Sequences with only a few notes occupy higher levels of the hierarchy and are embedded close to the origin. Musical pieces with more notes and hence a richer structure are located farther away at lower levels, such that the growth of the hierarchy aligns with the expansion of the hyperbolic space. Moreover, musical keys spread out in distinct directions from the origin, preserving their relations known from music theory and allow for semantic interpretation. For example, changing the key of a composition can be realized by a rotation in latent space, where the angle is dictating the difference between original and target key.

Exploiting this natural embedding becomes possible by a generalization of spherical interpolations (SLERP, Shoemake (1985)) to Riemannian manifolds. Our contribution allows for the first time to compute interpolation paths with respect to different levels of hierarchies in hyperbolic spaces, leading to smooth and meaningful transitions of keys and musical richness for music generation. In general, R-SLERP offers a volume-controlled interpolation scheme on Riemannian manifolds.

We empirically demonstrate the utility of hyperbolic music representations by comparing VAEs with hyperbolic and Euclidean latent spaces. To demonstrate the generalisability of the approach, we perform experiments on both symbolic and and raw waveform data. The results show that the hyperbolic model outperforms its Euclidean peers in terms of both interpretability and reconstruction performance.

## 2 RELATED WORK

Manifold-aware representations can be obtained by analysing the mappings between input and latent spaces for distortions. For example, Arvanitidis et al. (2018) propose a manifold-aware interpretation of latent spaces of VAEs by deriving a stochastic Riemannian metric from the encoder. Similarly, deterministic Riemannian metrics for measuring distances in latent spaces of VAEs and generative adversarial networks (GANs) can be derived from the generator models (Chen et al., 2018). While these approaches improve the interpretability of latent spaces, the spaces are intrinsically Euclidean during model training.

Alternatively, colleagues propose to utilize prior knowledge on non-Euclidean structures to embed data on specific (Riemannian) manifolds like Poincaré balls (Nickel & Kiela, 2017). A key publication by Nagano et al. (2019) introduces the Wrapped Normal distribution as a generalization of the Normal distribution to hyperbolic geometry, allowing to learn VAEs with hyperbolic latent spaces. Mathieu et al. (2019) and Ovinnikov (2019) develop alternative hyperbolic VAEs on the Poincaré ball model, which differ in the formulation of the optimization objective and both demonstrate the fit of hierarchical data to the geometry. Going beyond latent spaces with constant curvature, Skopek et al. (2020) propose VAEs on product manifolds, composed of component spaces with constant curvature. Building on these approaches, we develop a VAE that embeds music on a Poincaré ball. Moreover, we generalize SLERP to Riemannian manifolds to allow for interpolations that respect the organization of hierarchies in hyperbolic space, also giving rise to new applications of the previously proposed models.

VAEs are frequently used to learn smooth latent spaces for music (Fabius et al., 2015; Roberts et al., 2018) with the goal of obtaining interpretable and disentangled latent spaces. For example, Brunner et al. (2018) propose to train classifiers alongside a VAE that act on only a subspace of the latent representation to yield a disentangled encoding of the style of a composition. Consequentially, disentanglements of static and dynamic features (Li & Mandt, 2018), pitch and rhythm (Yang et al., 2019), or chords and texture (Wang et al., 2020b) have been studied with altered VAEs. While these approaches successfully disentangle certain attributes of music by supervision or tailored changes in model architecture, they all rely on Euclidean geometry. By contrast, we show in this paper that embedding music in hyperbolic space naturally yields interpretable representations of keys and musical richness without supervision and, hence, consider our approach orthogonal to this line of prior work.

Nevertheless, there already exist a few publications that deal with non-Euclidean representations of music. For example Nakashima et al. (2022) propose a hyperbolic VAE to embed sounds of single notes, played with different instruments, and Huang et al. (2023) propose a hyperbolic transformer

that makes use of hierarchical structures within individual pieces of music, such as the relations between beats, bars, and phrases. Chen et al. (2022) study a flat manifold as latent space by regularising its metric to be a scaled version of the Euclidean metric. While this improves the smoothness of the latent space, the space remains effectively Euclidean. By contrast, we focus on capturing non-Euclidean structures in music in form of hierarchies to reduce distortions of distances, ultimately aiming at interpretable latent spaces for music.

## 3 RIEMANNIAN GEOMETRY

In the following, foundational concepts of Riemannian geometry are briefly reviewed. For a more exhaustive description we confer to Lee (2018), which we also follow notation-wise.

A Riemannian manifold is a smooth, differentiable manifold $\mathcal{M}$ which is equipped with a Riemannian metric $\mathfrak{g}$, together forming the tuple $(\mathcal{M}, \mathfrak{g})$. At each point $z \in \mathcal{M}$ there exists a Euclidean space, called the tangent space $\mathcal{T}_z\mathcal{M}$, of the same dimensionality as the manifold that lies tangential to it. The Riemannian metric $\mathfrak{g}$ defines an inner product on each tangent space $\mathfrak{g}_z = \langle \cdot, \cdot \rangle_z : \mathcal{T}_z\mathcal{M} \times \mathcal{T}_z\mathcal{M} \to \mathbb{R}$. This inner product gives rise to the norm: $\|\cdot\|_z = \sqrt{\langle \cdot, \cdot \rangle_z}$.

**Distances and geodesics**  Assume a curve $\gamma(t)$ on $\mathcal{M}$, where $t \in [0, 1]$. For each point along the curve, its velocity is defined as $\gamma'(t)$. The norm of the velocity vector $\|\gamma'(t)\|_{\gamma(t)}$ can be integrated to calculate the length of the curve by

$$L(\gamma) = \int_0^1 \|\gamma'(t)\|_{\gamma(t)} \, dt. \tag{1}$$

The infimum of the lengths of all curves connecting two points $z$ and $y$ on $\mathcal{M}$ is considered their distance, formally defined as

$$\text{dist}(z, y) = \inf_\gamma L(\gamma) \qquad \text{with} \qquad \gamma(0) = z, \quad \gamma(1) = y. \tag{2}$$

Curves that locally minimize distances are called geodesics. While such curves can generally be defined with different parametrizations, geodesics are parametrized such that their speed $\|\gamma'(t)\|_{\gamma(t)}$ is constant. If all geodesics of a manifold can be extended to any $t \in \mathbb{R}$, the manifold is geodesically complete. All Riemannian manifolds considered in this work, i.e. the Poincaré ball, are known to be geodesically complete (Lee, 2018).

The infimum in equation 2 is always realized by some geodesic (Lee, 2018). Intuitively, geodesics can be seen as a generalization of straight lines to curved surfaces. In Euclidean space, where the curvature of space is zero, geodesics appear as straight lines in the conventional sense.

**Exponential and logarithmic map**  The exponential map $\exp_z$ defines a mapping of velocity vectors $v$ that live in the tangent space $\mathcal{T}_z\mathcal{M}$ to a corresponding point on the manifold. This is done by traveling along the unique geodesic that passes through $z$ at $t = 0$ with velocity $v$ for a unit time interval, denoted as $\gamma_{(z,v)}$:

$$\exp_z : \mathcal{T}_z\mathcal{M} \to \mathcal{M}, \quad v \mapsto \exp_z(v) = \gamma_{(z,v)}(1). \tag{3}$$

As the geodesic $\gamma_{(z,v)}$ has constant speed $\|\gamma'_{(z,v)}(t)\|_{\gamma_{(z,v)}(t)} = \|v\|_z \, \forall t$, the (Riemannian) distance between $z$ and $\exp_z(v)$ can be calculated as

$$\text{dist}(z, \exp_z(v)) = L(\gamma_{(z,v)}) = \int_0^1 \|\gamma'_{(z,v)}(t)\|_{\gamma_{(z,v)}(t)} \, dt = \int_0^1 \|v\|_z \, dt = \|v\|_z. \tag{4}$$

This shows that the exponential map preserves distances with respect to $z$ when mapping from $\mathcal{T}_z\mathcal{M}$ to $\mathcal{M}$.

The logarithmic map $\log_z(y)$ is the inverse of the exponential map. For two points $z$ and $y$ on the manifold, it calculates the velocity $v$ in tangent space $\mathcal{T}_z\mathcal{M}$ that the connecting geodesic $\gamma_{(z,v)}$ must have to reach $y$ in unit time:

$$\log_z = \exp_z^{-1} : \mathcal{M} \to \mathcal{T}_z\mathcal{M}, \quad y \mapsto \log_z(y) = \log_z^{-1}(y) = \gamma'_{(z,y)}(0). \tag{5}$$

As the exponential map, the logarithmic map preserves distances with respect to $z$.

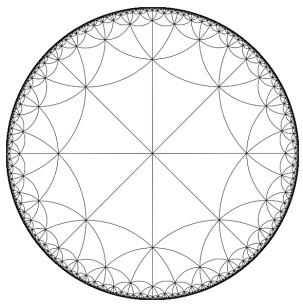

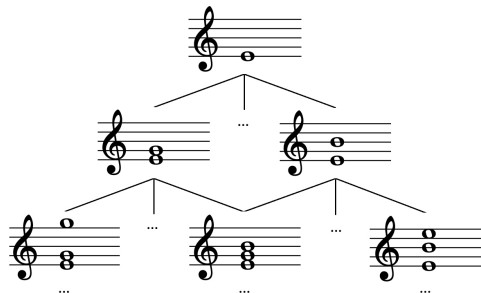

(a) Poincaré ball (with curvature $c < 0$)     (b) Hierarchy of note combinations

Figure 2: Left: Tessellation of the Poincaré ball of congruent triangles. Right: Hierarchical relations between notes of music.

**Parallel transport** Parallel transport is an operation which transports velocities $v$ from $\mathcal{T}_z\mathcal{M}$ to $\mathcal{T}_y\mathcal{M}$ while keeping the angle between $v$ and its transported variant $\tilde{v}$ constant. Technically, the parallel transport PT along a geodesic $\gamma_{(z,v)}$ is written as $\mathrm{PT}_{z\to y} : \mathcal{T}_z\mathcal{M} \to \mathcal{T}_y\mathcal{M}$. The length of a velocity vector $v \in \mathcal{T}_z\mathcal{M}$ is preserved under parallel transport, i.e. $\|v\|_z = \|\mathrm{PT}_{z\to y}(v)\|_y$. The inverse mapping of $\mathrm{PT}_{z\to y}$ transports velocities from $\mathcal{T}_y\mathcal{M}$ back to $\mathcal{T}_z\mathcal{M}$ and is denoted as $\mathrm{PT}_{z\to y}^{-1} = \mathrm{PT}_{y\to z}$.

**Poincaré ball** The most important invariant of Riemannian manifolds is the curvature $c$, which describes how the metric influences the space around a point. For example, values $c > 0$ contract the space and result in spherical geometry, $c = 0$ is the flat Euclidean case and $c < 0$ expands the space, yielding hyperbolic geometry. Different isometric models of hyperbolic geometry are available, the Poincaré ball is the one we utilise in this work. The Poincaré ball with $d$ dimensions and curvature $c$ is defined as $\mathbb{P}_c^d = (\mathcal{B}_c^d, \mathfrak{g}_{\mathbb{P}}^c(z))$. Here, $\mathcal{B}_c^d$ is the open ball $\{z \in \mathbb{R}^d \,|\, \|z\|_2 < R\}$ with radius $R = 1/\sqrt{c}$, the Riemannian metric $\mathfrak{g}_{\mathbb{P}}^c(z)$ is defined as

$$\mathfrak{g}_{\mathbb{P}}^c(z) = (\lambda_z^{\mathbb{P}})^2 \mathfrak{g}_{\mathbb{E}} \qquad \text{with} \qquad \lambda_z^{\mathbb{P}} = \frac{2}{1 + c\|z\|_2^2} \tag{6}$$

where $\mathfrak{g}_{\mathbb{E}}$ is the Euclidean metric (Mathieu et al., 2019). The exponential map, the logarithmic map and distance function for the Poincaré ball are given in Appendix A.

## 4 LEARNING TO REPRESENT MUSIC

### 4.1 MOTIVATION

Hierarchically structured data is commonly represented in form of trees. When such a tree is embedded in a metric space, the nodes and leafs become naturally denser with increasing levels of hierarchy. Embedding a tree in a Euclidean space renders the leafs difficult to separate if they are deep enough in the tree because the distance between them decreases, while the space does not expand. By contrast, in hyperbolic geometry, the space expands under the influence of the negative curvature and directly counteracts large tree widths by providing more space to deeper levels and, hence, leafs become easier separable (Nickel & Kiela, 2017). Figure 2a visualises this growth by showing a tessellation of the Poincaré ball of congruent triangles: The space expands with respect to the distance to the origin and culminates in infinite distance at the boundary. We now show that embedding music in a Poincaré ball offers a direct (semantic) interpretation of the representation.

Musical compositions are not randomly structured, they adhere to different paradigms and ideas about structures and tonal combinations. Clearly, not all notes sound harmonic together.[1] Choosing one note influences the choice of other notes being played simultaneously. The resulting 'harmonic' hierarchy is visualized in Figure 2b and resembles a tree-like structure as the choice of the third note also depends on the former two and so forth. Playing more notes together while staying harmonic

---

[1] We do not differentiate between harmonic consonance or dissonance and simply refer to harmonic.

constitutes an increase in musical richness, which we, together with the key of the music, consider essential ingredients of latent representations of music. In Figure 1 and 2b, the musical richness increases with the distance to the origin (Figure 1) and vice versa with increasing tree depths (Figure 2b). Hence, there is a direct relation between $n$-note chords and their embedding in a Poincaré ball.

## 4.2 Learning music VAEs with Poincaré balls

Variational autoencoders (Kingma & Welling, 2014; Rezende et al., 2014) aim at modeling data generating processes in form of a prior $p(\boldsymbol{z})$ and a conditional distribution $p(\boldsymbol{x}|\boldsymbol{z})$ to generate instances $\boldsymbol{x}$ given their latent representation $\boldsymbol{z}$. The encoder-decoder architecture learns the variational posterior $q(\boldsymbol{z}|\boldsymbol{x})$ (encoder) and $p(\boldsymbol{x}|\boldsymbol{z})$ (decoder) by optimizing the ELBO criterion, a lower bound of the marginal log-likelihood.

In conventional VAEs, the latent space and all computations and distributions are assumed to be Euclidean. Following our argumentation, we now sketch how to define a VAE with a hyperbolic latent space for hierarchical structures such as music (Mathieu et al., 2019; Nagano et al., 2019). To parametrize the variational posterior, the encoder needs to output a velocity $\boldsymbol{v_\mu}$ at $\mathcal{T}_0\mathcal{M}$ leading to a mean $\mu$ on $\mathcal{M}$ along with a variance $\sigma$. We denote the origin on the manifold $\mathcal{M}$ by a subscript $0$. Sampling can then be performed via a Wrapped Normal distribution (Nagano et al., 2019).

Samples are drawn from $\mathcal{WN}(\mu, \sigma)$ as follows: A velocity $\boldsymbol{v}_\sigma$ is sampled from a Gaussian defined in the tangent space at the origin. This sampled velocity is transported into the tangent space at the mean $\mu$ of the distribution via $\mathrm{PT}_{0\to\mu}(\boldsymbol{v}_\sigma)$, resulting in $\tilde{\boldsymbol{v}}_\sigma$. Finally, the transported velocity is mapped onto the manifold via $\exp_\mu(\tilde{\boldsymbol{v}}_\sigma)$ to obtain the sample $\boldsymbol{z}$. Summarizing, this process can be written as

$$\boldsymbol{v}_\sigma \sim \mathcal{N}(0, \sigma) \in \mathcal{T}_o\mathcal{M}, \quad \tilde{\boldsymbol{v}}_\sigma = \mathrm{PT}_{0\to\mu}(\boldsymbol{v}_\sigma) \in \mathcal{T}_\mu\mathcal{M}, \quad \boldsymbol{z} = \exp_\mu(\tilde{\boldsymbol{v}}_\sigma) \in \mathcal{M}. \quad (7)$$

The result of $\log_0(\boldsymbol{z})$ is then decoded in standard fashion to produce $\hat{\boldsymbol{x}}$.

The hyperbolic VAE with Poincaré ball is optimized by minimizing an adapted version of the ELBO that approximates the KL-divergence between variational posterior and prior as an unbiased Monte Carlo estimate, calculated as

$$L(\boldsymbol{x}, \hat{\boldsymbol{x}}, \mu, \sigma) = \mathrm{RE}(\boldsymbol{x}, \boldsymbol{y}) + \beta \left( \frac{1}{N} \sum_{j=1}^{N} \log q(z_j|\boldsymbol{x}) - \log p(z_j) \right) \quad (8)$$

with

$$q(\boldsymbol{z}|\boldsymbol{x}) = \mathcal{WN}(\mu, \sigma \cdot I), \quad p(\boldsymbol{z}) = \mathcal{WN}(0, I), \quad z_j \sim q(\boldsymbol{z}|\boldsymbol{x}) \quad \text{for} \quad j = 1, ..., N \quad (9)$$

where RE is the reconstruction error, $q(\boldsymbol{z}|\boldsymbol{x})$ is the variational posterior, and $p(\boldsymbol{z})$ is the prior. A hyperparameter $\beta$ controls the scaling of the KL divergence term to balance reconstruction quality and latent space regularization, following the $\beta$-VAE approach (Higgins et al., 2017). Appendix B shows the $\log$-probability of the Wrapped Normal.

To address the sequential structure of music, our encoder network is a bidirectional LSTM with two layers (Hochreiter & Schmidhuber, 1997; Schuster & Paliwal, 1997), following the architecture from Roberts et al. (2018) and the decoder a two-layer unidirectional LSTM.

## 4.3 Riemannnian SLERP

Recall that distance to the origin is proportional to hierarchy levels in embeddings on Poincaré balls. We now aim to generate interpolations on the ball that change linearly in the distance to the origin along the interpolation path. That is, two instances on the ball are interpolated in a way that the hierarchy level of the interpolation curve changes at a constant rate and is guaranteed to stay within the hierarchy levels of its start and end point.

We propose to construct interpolation paths in the tangent space at the origin of the Poincaré ball. Latent representations of the original instances $\boldsymbol{z_1}, \boldsymbol{z_2} \in \mathbb{P}_c^d$ are mapped to velocities $\boldsymbol{v_1}, \boldsymbol{v_2} \in \mathcal{T}_0\mathbb{P}_c^d$, which are used to compute the interpolated velocities $\boldsymbol{v}_{\mathrm{int}}(t) \in \mathcal{T}_0\mathbb{P}_c^d$. We map the velocities back onto the Poincaré ball to obtain the corresponding interpolation points $\boldsymbol{z}_{\mathrm{int}}(t) \in \mathbb{P}_c^d$. The map

between $\mathbb{P}_c^d$ and $\mathcal{T}_\mathbf{0}\mathbb{P}_c^d$ is carried out using exponential and logarithmic maps (cf. Section 3) that preserve distances with respect to their anchor points. Consequentially, the geodesic distances of the interpolated points $\boldsymbol{z}_\text{int}(t)$ to the origin will change linearly with $t$, given that such behaviour holds for $\boldsymbol{v}_\text{int}(t)$.

In general, interpolations by straight lines in Euclidean space or geodesics on manifolds fail to preserve the norm towards the origin. We thus resort to SLERP (Shoemake, 1985) to render the norm of the rotating vectors constant. Therefore, interpolations between two velocities $\boldsymbol{v}_1, \boldsymbol{v}_2 \in \mathcal{T}_\mathbf{0}\mathbb{P}_c^d$ are calculated as follows. The direction of the interpolated velocity $\hat{\boldsymbol{v}}_\text{int}(t)$ is determined independently of its magnitude by normalizing $\boldsymbol{v_1}$ and $\boldsymbol{v_2}$ to unit length. Given the two unit vectors $\hat{\boldsymbol{v}}_1$ and $\hat{\boldsymbol{v}}_2$, SLERP is defined as

$$\text{SLERP}(\hat{\boldsymbol{v}}_1, \hat{\boldsymbol{v}}_2; t) = \frac{\sin((1-t)\theta)}{\sin(\theta)}\hat{\boldsymbol{v}}_1 + \frac{\sin(t\theta)}{\sin(\theta)}\hat{\boldsymbol{v}}_2 \quad \text{with} \quad \theta = \arccos(\hat{\boldsymbol{v}}_1 \cdot \hat{\boldsymbol{v}}_2). \tag{10}$$

We now generalise this approach to Riemannian manifolds and add a scaling component. The resulting Riemannian spherical linear interpolation (R-SLERP) is derived as follows.

Firstly, note that changing $\hat{\boldsymbol{v}}_1$ and $\hat{\boldsymbol{v}}_2$ from a Euclidean space $\mathbb{E}$ to a Riemannian manifold $\mathcal{M}$ has several implications. Viewing $\mathbb{E}$ from differential geometry, the tangent space $\mathcal{T}\mathcal{M}$ does not depend on $\hat{\boldsymbol{v}}_1$ or $\hat{\boldsymbol{v}}_2$, but is globally identified with $\mathbb{E}$. However, on Riemannian manifolds, $\mathcal{T}_\boldsymbol{z}\mathcal{M}$ depends on $\boldsymbol{z}$ and the Riemannian volume element $\omega_\mathfrak{g}$ changes accordingly. As this change would displace linear interpolations, we need to control the volume element.

The Riemannian volume form $\omega_\mathfrak{g}$, given in local coordinates of a manifold $\mathcal{M}$ of finite dimension $d$, is defined as

$$\omega_\mathfrak{g} = \sqrt{\det(\mathfrak{g}_{ij})}d\boldsymbol{z}^1 \wedge \ldots \wedge d\boldsymbol{z}^d, \tag{11}$$

where $\mathfrak{g}_{ij}$ are the components of $\mathfrak{g}$ in these co-ordinates. A complete, simply connected Riemannian manifold with constant sectional curvature is isometric to one of the model spaces, i.e. $\mathbb{R}, \mathbb{S}, \mathbb{P}$, due to the Killing-Hopf theorem (cf. 12.4 Lee (2018)). Recall that the model spaces are isotropic, furthermore, we can assume an origin.

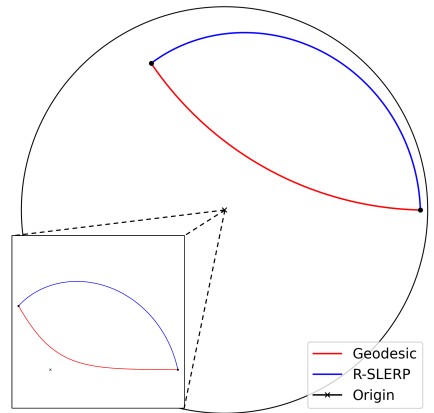

Figure 3: Interpolation paths obtained with R-SLERP compared to interpolation along geodesic on $\mathbb{P}^2$ and in $\mathcal{T}_\mathbf{0}\mathbb{P}^2$.

Thus, we can focus on $\text{dist}(\mathbf{0}, \boldsymbol{z})$ to control $\omega_\mathfrak{g}$ and do not need to care about the exact value of $\omega_\mathfrak{g}$ to maintain the exact positions of interpolated values. Note that the norm $\|\boldsymbol{v_z}\| \in \mathcal{T}_\mathbf{0}\mathcal{M}$ can be used to control the distance of $\boldsymbol{z}$ to the origin since $\|\boldsymbol{v_z}\| = \text{dist}(\mathbf{0}, \exp_\mathbf{0}(\boldsymbol{v_z})) = \text{dist}(\mathbf{0}, \boldsymbol{z})$ (see equation 4).

We now make use of this distance preserving property of the exponential map to map from $\mathcal{T}_\boldsymbol{z}\mathcal{M} \mapsto \mathcal{M}$. That is, we draw the calculation of the interpolation into $\mathcal{T}_\mathbf{0}\mathcal{M}$ and obtain $\boldsymbol{v_i} = \log_\mathbf{0}(\boldsymbol{z_i})$ for $i = 1, 2$. The resulting velocities $\boldsymbol{v_1}, \boldsymbol{v_2} \in \mathcal{T}_\mathbf{0}\mathcal{M}$ are now used to compute the interpolated velocities $\boldsymbol{v}_\text{int}(t) \in \mathcal{T}_\mathbf{0}\mathcal{M}$.

Normalising $\boldsymbol{v_1}, \boldsymbol{v_2}$ to unit length allows the application of vanilla SLERP in $\mathcal{T}_\mathbf{0}\mathcal{M}$, resulting in $\hat{\boldsymbol{v}}_\text{int}(t)$. As vanilla SLERP uses unit vectors, we calculate

$$\alpha(t) = (1-t)\|\boldsymbol{v}_1\| + t\|\boldsymbol{v}_2\| \tag{12}$$

to be able to scale linearly between the different-norm vectors. The rescaled $\boldsymbol{v}_\text{int}(t) = \alpha(t)\hat{\boldsymbol{v}}_\text{int}(t)$ is then pushed back onto the manifold $\mathcal{M}$, resulting in the interpolated point $\boldsymbol{z}_\text{int}(t)$ given by

$$\boldsymbol{z}_\text{int}(t) = \exp_\mathbf{0}(\boldsymbol{v}_\text{int}(t)). \tag{13}$$

Algorithm 1 summarizes the operations for R-SLERP. The blue curves in Figure 3 exemplarily show the interpolation paths obtained by the proposed method on a Poincaré ball and in the tangent space at its origin, respectively.

---

**Algorithm 1** Riemannian-SLERP (R-SLERP)

---

**Require:** latent representations $\boldsymbol{z}_1, \boldsymbol{z}_2 \in \mathbb{P}_c^d$, parameter $t \in [0, 1]$
1: $\boldsymbol{v}_1 \in \mathcal{T}_0\mathbb{P}_c^d \leftarrow \log_{\mathbf{0}}(\boldsymbol{z}_1), \quad \boldsymbol{v}_2 \in \mathcal{T}_0\mathbb{P}_c^d \leftarrow \log_{\mathbf{0}}(\boldsymbol{z}_2)$
2: $\hat{\boldsymbol{v}}_1 \leftarrow \frac{\boldsymbol{v}_1}{\|\boldsymbol{v}_1\|}, \quad \hat{\boldsymbol{v}}_2 \leftarrow \frac{\boldsymbol{v}_2}{\|\boldsymbol{v}_2\|}$
3: $\hat{\boldsymbol{v}}_{\text{int}}(t) \leftarrow \text{SLERP}(\hat{\boldsymbol{v}}_1, \hat{\boldsymbol{v}}_2; t)$
4: $\alpha(t) \leftarrow (1-t)\|\boldsymbol{v}_1\| + t\|\boldsymbol{v}_2\|$
5: $\boldsymbol{v}_{\text{int}}(t) \leftarrow \alpha(t)\hat{\boldsymbol{v}}_{\text{int}}(t)$
6: $\boldsymbol{z}_{\text{int}}(t) \in \mathbb{P}_c^d \leftarrow \exp_{\mathbf{0}}(\boldsymbol{v}_{\text{int}}(t))$
7: **return** $\boldsymbol{z}_{\text{int}}(t)$

---

## 5 EXPERIMENTS

**Data**  We run experiments on three datasets with different modalities: For symbolic music we use the POP909 (Wang et al., 2020a) and MIDICAPS (Melechovský et al., 2024) datasets, for raw waveform the MAESTRO (Hawthorne et al., 2019) dataset.

POP909 consists of 909 piano versions of popular songs in MIDI format with a total duration of 60 hours, the annotations include musical keys. MIDICAPS is based on the Lakh MIDI data (Raffel, 2016) and consists of 168,407 MIDI files. MAESTRO (Hawthorne et al., 2019) contains 1,276 recorded piano performances with a total duration of 199 hours in raw waveform.

Both symbolic music datasets are split into 70% training, 15% validation and 15% test sets. The data is preprocessed from piano rolls into pseudo wave form, following Prang & Esling (2021). We use the signal-like representation for VAE inputs and outputs and resort to piano rolls for evaluations and visualizations. For MAESTRO, we transform the raw audio into Mel spectrograms and we use the original train, validation and test splits. Confer Appendix C.1 for details.

**Models**  Recall that prior work on VAEs for music primarily focuses on optimized architectures (Roberts et al., 2018; Caillon & Esling, 2021) or disentanglements of certain characteristics (Brunner et al., 2018; Li & Mandt, 2018; Yang et al., 2019; Wang et al., 2020b), while relying entirely on Euclidean geometry. Chen et al. (2022) propose a regularized Riemannian metric to approximate a scaled Euclidean geometry, which effectively yields a flat latent space with improved smoothness. As we aim at comparing hyperbolic against Euclidean geometry for representing music, all of these previously proposed contributions are orthogonal to our study and may further improve results.

To have a principled setup, we compare of a VAE with a Poincaré ball as latent space ($\mathbb{P}$-VAE) with a VAE operating on Euclidean geometry ($\mathbb{E}$-VAE). As the implementations of VAEs on Riemannian manifolds naturally differ from the default case, we additionally implement a standard Euclidean VAE (Kingma & Welling, 2014) and compare it to the $\mathbb{E}$-VAE on POP909 across ten runs. The performances do not differ significantly, which is expected, as they are mathematically equivalent.

We use latent spaces of dimensionality $128$ for the experiments on POP909 and MAESTRO, and $512$ for MIDICAPS. For hyperparameters that are integral for the effect of the latent space geometry, concerning $\beta$-values and curvature, grid searches are conducted, cf. Appendix C.2.

**Results**  We evaluate reconstruction performance on the test splits of all datasets over ten repetitions. Table 1 and 2 show the results. Values in bold face indicate significant results according to Welch's t-test at $p < 0.05$.

The $\mathbb{P}$-VAE outperforms the $\mathbb{E}$-VAE significantly on POP909 (F1, $p < 0.001$), MIDICAPS (F1, $p = 0.046$) and MAESTRO (MSE, $p < 0.001$). These results demonstrate impressively that hyperbolic latent spaces capture the internal structures of music data more effectively than Euclidean geometry. This indicates that the match of hyperbolic geometry with music holds for symbolic music and raw audio, and irrespective of the size of data and latent space.

In terms of computational efficiency, utilising a hyperbolic latent space increases the runtime per epoch by approximately 7-40%. Further analysis of training times and hyperparameter sensitivities are available in Appendix C.3 and C.2.

Table 1: Reconstruction performance on symbolic music. We report means and standard deviations.

|  | Model | F1-score (%) | Precision (%) | Recall (%) |
|---|---|---|---|---|
| POP909 | $\mathbb{E}$-VAE | $88.76_{\pm 0.18}$ | $90.88_{\pm 0.14}$ | $86.73_{\pm 0.25}$ |
|  | $\mathbb{P}$-VAE | $\mathbf{90.90_{\pm 0.06}}$ | $\mathbf{92.15_{\pm 0.11}}$ | $\mathbf{89.68_{\pm 0.08}}$ |
| MIDICAPS | $\mathbb{E}$-VAE | $97.68_{\pm 0.05}$ | $98.51_{\pm 0.04}$ | $\mathbf{96.86_{\pm 0.06}}$ |
|  | $\mathbb{P}$-VAE | $\mathbf{97.74_{\pm 0.08}}$ | $\mathbf{98.82_{\pm 0.05}}$ | $96.68_{\pm 0.11}$ |

**UMAP projection** To show the natural organization of musical keys in hyperbolic space, latent representations of the $\mathbb{P}$-VAE are visualized with UMAP (McInnes & Healy, 2018). We sample 2,000 instances from the POP909 test set and use their embeddings of the best performing model as inputs. The set of musical keys of these instances consists of every second major key in the circle of fifths (cf. Figure 4b), implying that the groups of notes associated with the keys differ by at least two notes. Embedding similarities are calculated as geodesic distances directly on the manifold.

Table 2: Reconstruction performance on MAESTRO.

| Model | MSE |
|---|---|
| $\mathbb{E}$-VAE | $27.60_{\pm 0.20}$ |
| $\mathbb{P}$-VAE | $\mathbf{25.13_{\pm 0.11}}$ |

Figure 4a shows that the keys are well separated in the hyperbolic latent space. Interestingly, the ordering of keys matches exactly their order in the circle of fifths. It becomes evident that the model learns the dependencies between keys known from music theory without supervision, meaning that the relations between musical pieces which are based on different sets of notes are preserved. The circular arrangement of keys is centered at the origin of the Poincaré ball, suggesting that R-SLERP enables meaningful transitions of keys, as the method is based on vector rotation around the origin.

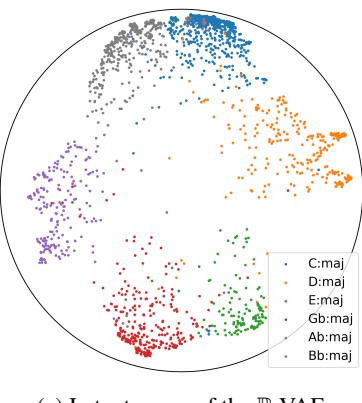
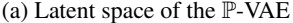
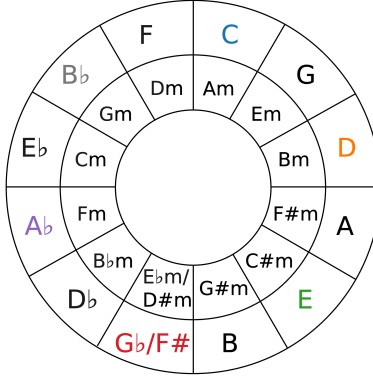

(a) Latent space of the $\mathbb{P}$-VAE

(b) Circle of fifths

Figure 4: Left: UMAP visualization showing the latent space of the $\mathbb{P}$-VAE, based on instances with major keys from the set {C, D, E, Gb, Ab, Bb} (mirrored and rotated). Right: The musical circle of fifths. The keys are perfectly aligned with the naturally learnt structure of the model.

**Interpolation with R-SLERP** To illustrate the hierarchical organization of music in hyperbolic latent spaces, interpolations between instances from POP909 are performed using the best performing $\mathbb{P}$-VAE model. We compare three different interpolations: (a) The piano rolls in Figure 5a are computed using our R-SLERP as described in Section 4.3. (b) For comparison, we interpolate along the connecting geodesic, the canonical approach to interpolation on manifolds (Arvanitidis et al., 2018; Chen et al., 2018). (c) Finally, the encoding of musical richness as distance to the origin is demonstrated through an additional interpolation from the origin towards an instance.

The geodesic interpolation in Figure 5b decreases the richness of the piano rolls towards the midpoint of the interpolation path that is only scarcely occupied by notes, corresponding to a higher level of hierarchy. Geodesics on the Poincaré ball are curved inwards, thus being unable to consistently vary hierarchy levels. Complementing this observation, the interpolation from origin to instance in Figure

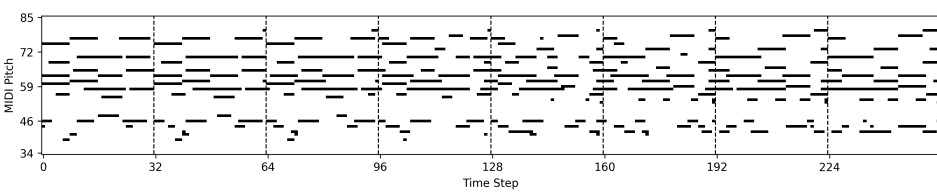

(a) Interpolation between two instances using R-SLERP.

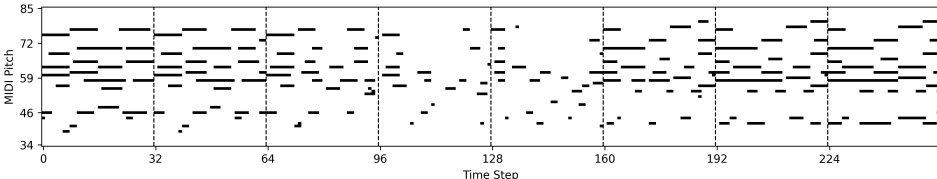

(b) Interpolation between two instances along geodesic.

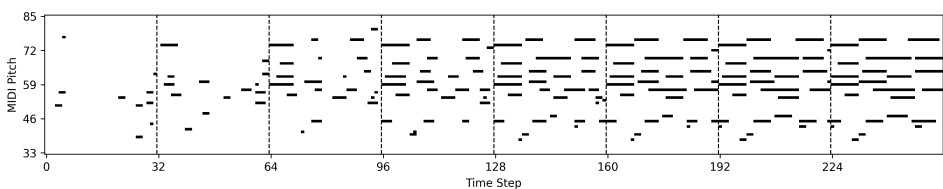

(c) Interpolation from origin to instance along geodesic.

Figure 5: Exemplary interpolations. Vertical lines mark the beginning of new piano rolls.

5c shows an increasing musical richness over time, whereby the origin corresponds to the almost 'empty' song. In contrast, the piano rolls obtained using R-SLERP in Figure 5a are distinguished by consistent musical richness along the entire interpolation path. By design, R-SLERP is guaranteed to stay within the hierarchical levels of the two instances and decreases/increases complexity linearly along the path, thus constituting an appropriate interpolation scheme for music.

## 6 CONCLUSION

Utilising hyperbolic geometry, we proposed to design latent spaces that explicitly reflect hierarchies in music. We generalized spherical linear interpolation to Riemannian manifolds to allow for meaningful interpolations. Empirically, we observed that hyperbolic music representations outperform Euclidean counterparts in terms of reconstruction and interpretation.

While we demonstrated our approach on both symbolic and raw waveform data, the benefit for other datasets (e.g., recordings with multiple instruments or singing) needs yet to be confirmed. Another limitation is the exploration of downstream tasks, which we left for future work in order to maintain a clear focus and principled approach. As the increase in structure in the latent space already improves reconstruction, we expect excellent performances of hyperbolic approaches in a variety of tasks beyond music generation. The hyperbolic embeddings capture the internal structure of music very well, rendering our approach a natural choice for applications that are based on similarities such as classification, recommendation, retrieval, or plagiarism detection.

In the field of music generation, powerful foundational models are readily available that are reshaping the process of music creation. Our work provides a first step towards reclaiming explicit control. We provide a building block for future model architectures which increases the internal representation quality, allowing for more interpretability and thus interactivity with the model. Time will tell whether this enhanced control allows for more human-model interaction to foster the creative process instead of replacing it.

## REPRODUCIBILITY STATEMENT

All experiments and models are described in the main text in Section 5, additional details are given in Appendix C. To enable the reproducibility of results, the experiments only involve datasets which are publicly available. The source code will be released upon publication. If further questions arise, the authors are available via e-mail.

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

## A  OPERATORS ON THE POINCARÉ BALL

The Poincaré ball $\mathbb{P}^c$ of dimension $d$ is a model of hyperbolic geometry. This geometry is especially useful for embedding data with hierarchical structures, as demonstrated by Mathieu et al. (2019). Due to the property of the space expanding when influenced by negative curvature, data structures do not exhibit the problem of becoming increasingly dense. The metric tensor of the Poincaré ball is given by:

$$\mathfrak{g}_{\mathbb{P}}^c(\boldsymbol{z}) = (\lambda_{\boldsymbol{z}}^c)^2 \, \mathfrak{g}_{\mathbb{E}}(\boldsymbol{z}), \quad \lambda_{\boldsymbol{z}}^c = \frac{2}{1 - c\|\boldsymbol{z}\|^2}, \tag{14}$$

where $\lambda_{\boldsymbol{z}}^c$ is the conformal factor and $\mathfrak{g}_{\mathbb{E}}$ is the Euclidean metric tensor. Please note that these formulas expect $c$ to be a positive value, even though it is a negative curvature. We choose to adhere to Mathieu et al. (2019) for consistency and to not add to the already extensive available inventory of formulas.

The exponential map is given by:

$$\exp_{\boldsymbol{z}}^c(\boldsymbol{v}) = \mu \oplus \left( \tanh \sqrt{c} \frac{\lambda_{\boldsymbol{z}}^c \|\boldsymbol{v}\|}{2} \frac{\boldsymbol{v}}{\sqrt{c}\|\boldsymbol{v}\|} \right) \tag{15}$$

where $\oplus$ is denoting the Möbius addition (Ungar, 2009). The Möbius addition composes velocities, which is possible in spaces that are equipped with a gyrovector-structure, which hyperbolic geometry does. The inverse operation for the exponential map, the logarithm map is given by:

$$\log_{\boldsymbol{z}}^c(\boldsymbol{y}) = \frac{2}{\sqrt{c}\lambda_{\boldsymbol{z}}^c} \tanh^{-1}\left( \sqrt{c}\| -\boldsymbol{z} \oplus_c \boldsymbol{y} \| \right) \frac{-\boldsymbol{z} \oplus_c \boldsymbol{y}}{\| -\boldsymbol{z} \oplus_c \boldsymbol{y} \|} \tag{16}$$

Distance on the Poincaré ball is given by:

$$d^c(\boldsymbol{z}, \boldsymbol{y}) = \frac{1}{\sqrt{c}} \cosh^{-1}\left( 1 + 2c \frac{\|\boldsymbol{z} - \boldsymbol{y}\|^2}{(1 - c\|\boldsymbol{z}\|^2)(1 - c\|\boldsymbol{y}\|^2)} \right) \tag{17}$$

## B  LOG-PROBABILITY OF THE WRAPPED NORMAL DISTRIBUTION

Let $f(\boldsymbol{v}) = \exp_\mu(\boldsymbol{v}) \circ \mathrm{PT}_{0 \to \mu}(\boldsymbol{v})$ be the composed function mapping velocities sampled at the origin onto the manifold. Then, the log-probability of a sample $\boldsymbol{z} \in \mathcal{M}$ can be calculated as

$$\log p(\boldsymbol{z}) = \log p(\boldsymbol{v}) - \log \det(J_f(\boldsymbol{v})) \tag{18}$$

whereby the velocity $\boldsymbol{v} \in \mathcal{T}_0\mathcal{M}$, corresponding to the sample, can be obtained by applying the inverse of $f$ which is $f^{-1} = \mathrm{PT}_{\mu \to 0} \circ \log_\mu$ to $\boldsymbol{z}$ (Nagano et al., 2019). Conceptually, $\log p(\boldsymbol{v})$ is the log-probability of $\boldsymbol{v}$ with respect to the Gaussian defined in $\mathcal{T}_0\mathcal{M}$, and $\log \det(J_f(\boldsymbol{v}))$ is the log-determinant of the Jacobian of $f$, accounting for the effect of $f$ on the probability density (Pennec, 2006).

## C  EXPERIMENTAL DETAILS

### C.1  DATA PREPROCESSING

**Symbolic music**  We convert the MIDI files into binary piano rolls by sampling the notes at 16 equidistant points in time per bar, which effectively discretises the time dimension and normalises tempo. Binary piano rolls are matrices where rows indicate pitch and columns indicate time. In a binary piano roll, every pitch can only be on or off at a time. The resulting piano rolls are split into sequences with a length of two bars using a sliding window, resulting in about 70,000 instances for POP909, and 1.5 million for MIDICAPS.

While the piano roll representation is intuitive and interpretable, it is usually very sparse because only a fraction of the available pitches are played at the same time. Thus, we transform the piano rolls into artificial sound waves, the so-called signal-like representation (Prang & Esling, 2021). The transformation is invertible and yields spectrograms by mapping pitches to arbitrary frequencies, from which signals can be obtained via short-time Fourier synthesis (Rabiner & Schafer, 2007). The

representations are visualized in Figure 6. We use the signal-like representation for VAE inputs and outputs and resort to piano rolls for evaluations and visualisations.

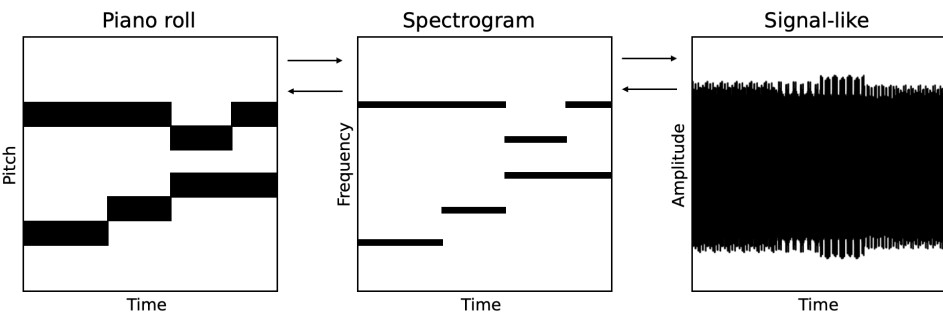

Figure 6: Symbolic music representations

**Raw audio**   For the experiments on recorded audio, we downsample the raw waveform to 22,050 Hz and divide each piece into segments with a length of 4 seconds using a hop size of 2 seconds, yielding approximately 360,000 instances for MAESTRO. Further, the snippets are converted into Mel spectrograms with 173 time steps and 128 Mel bands per segment, whereby values are clipped to lie in the range of -80 to 0 decibel. For VAE inputs and outputs, the Mel spectrograms are normalised to range $(0, 1)$, evaluations are performed on the unnormalised spectrograms.

## C.2   MODEL CONFIGURATIONS

**Model selection**   Model hyperparameters that are integral for the effect of the latent space geometry on model performance are optimized via grid searches for each model and dataset. For the $\mathbb{E}$-VAE, different weights of the KL term in the loss function $\beta \in \{0, 0.001, 0.01, 0.1, 1.0\}$ are considered. Also for the $\mathbb{P}$-VAE, $\beta$ is varied using the same search grid. In addition, different curvature values of the Poincaré ball $c \in \{-0.75, -1.0, -1.5\}$ are considered on POP909 and MAESTRO. For the experiments on MIDICAPS, the search grid is extended by curvature values $\{-0.1, -0.25, -0.5\}$. Table 3 displays the hyperparameter combinations yielding the best performances on the validation splits.

Table 3: Best hyperparameter configurations

|        |        | POP909 | MIDICAPS | MAESTRO |
|--------|--------|--------|----------|---------|
| $\mathbb{E}$-VAE | $\beta$ | 0.01 | 0.1 | 0.01 |
| $\mathbb{P}$-VAE | $\beta$ | 0.01 | 0.01 | 0.001 |
|        | $c$ | $-0.75$ | $-0.1$ | $-1.5$ |

**Hyperparameter sensitivity**   The hyperparameter sensitivities of the $\mathbb{E}$-VAE and $\mathbb{P}$-VAE on POP909 are shown in figure 7. One can observe that both models are relatively robust towards moderate changes in $\beta$, however, large values seem to degrade performance. Similarly, high curvature values decrease the performance of the $\mathbb{P}$-VAE, in particular if $\beta$ is suboptimally chosen.

## C.3   TRAINING

**Training configurations**   Due to the choice of input representations, mean squared error is used as the reconstruction error term in the training objectives. Model trainings are performed using early stopping with a patience of five epochs and a maximum number of 50

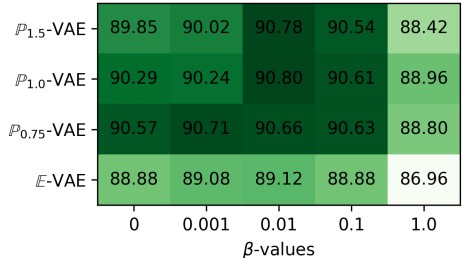

Figure 7: Hyperparameter sensitivities.

epochs. For optimization, the Adam optimizer (Kingma & Ba, 2015) is used with an initial learning rate of 0.0001 which is reduced every 10 epochs by a factor of 0.5. Further, batch sizes of 32, 2048, and 512 are chosen for trainings on POP909, MIDICAPS, and MAESTRO, respectively. A weight decay of $1 \times 10^{-5}$ is used. For the KL divergence estimation via MC sampling, 32 samples are drawn per instance.

**Runtimes**   To compare the computational efficiency of the $\mathbb{E}$-VAE and $\mathbb{P}$-VAE, we track training durations over ten repetitions per model and dataset on NVIDIA A100 GPUs. Table 4 shows the runtimes in seconds per epoch. We observe that the $\mathbb{P}$-VAE requires 7-40% more computation time, depending on the model and training configurations.

Table 4: Runtimes in seconds per epoch. We report means and standard deviations.

|  | POP909 | MIDICAPS | MAESTRO |
|---|---|---|---|
| $\mathbb{E}$-VAE | $98.02_{\pm 1.20}$ | $507.73_{\pm 14.58}$ | $263.03_{\pm 2.16}$ |
| $\mathbb{P}$-VAE | $137.23_{\pm 4.39}$ | $556.87_{\pm 7.67}$ | $280.52_{\pm 2.59}$ |

## C.4   IMPLEMENTATION DETAILS

**Packages**   The VAE models are implemented using the deep learning library PyTorch (Paszke et al., 2019). As a basis for the implementation of operations on manifolds, Geoopt (Kochurov et al., 2020) is used. MIDI files are processed using Pretty_midi (Raffel & Ellis, 2014). The library Librosa (McFee et al., 2015) is used to perform short-time Fourier transform and synthesis, and to produce Mel spectrograms.

**Numerical stability**   In the implementations of VAEs on manifolds, numerical stability is improved with the following two measures. At first, the log-variances outputted by the encoder networks are clipped at a minimum of -18 and a maximum of 0, which stabilizes the estimation of KL divergences. This should not limit the models significantly, as extremely small values below -18 cause very large KL divergences and thus are unlikely to be beneficial. Further, when log-variances increase to values greater than 0, the KL divergence increases while reconstructions get more noisy, which is also unlikely to have a positive effect on the loss. As a second measure, the exponential map of the Poincaré ball is implemented such that a minimum distance of outputted points to the border of the manifold can be enforced to prevent situations in which numerical imprecisions cause points to be mapped directly onto the border of the manifold where distances go to infinity.

