# OpenReview forum: "Hyperbolic Music Representations"
_ICLR.cc/2026/Conference — ICLR 2026 Conference Withdrawn Submission_

### Official Review · Reviewer_nH73 · 2025-10-31

**Soundness:** 3
**Presentation:** 4
**Contribution:** 2
**Rating:** 2
**Confidence:** 3

**Summary:**

The paper proposes learning hyperbolic representation spaces for music using a VAE with a Poncare ball as the latent space, capturing the hierarchical nature of musical structures (e.g., keys, chords, and variations) more effectively than Euclidean geometry.

It introduces a novel Riemannian Spherical Linear Interpolation (R-SLERP) method, enabling smooth and meaningful interpolations between musical pieces by respecting hierarchical distances and curvature in hyperbolic space.

Experiments on MIDI and raw audio datasets show that the proposed P-VAE achieves better interpretability and reconstruction performance than standard Euclidean VAEs.

**Strengths:**

The use of hyperbolic geometry allows the model to naturally capture the hierarchical structure of musical concepts (such as notes -> chords -> progressions -> full pieces), which is difficult for Euclidean embeddings to represent effectively.

By introducing R-SLERP in the hyperbolic latent space, the model produces smoother and more musically coherent transitions between pieces, leading to more interpretable and meaningful latent representations.

**Weaknesses:**

Hyperbolic latent spaces and VAEs have been explored in prior work (e.g., Nagano et al., 2019; Mathieu et al., 2019). The paper’s contribution mainly lies in applying it to music and introducing R-SLERP, so the core idea of hyperbolic representation is not entirely new.

While the paper argues that music is hierarchical, the intuition behind why hyperbolic embeddings are particularly suitable for musical structures (beyond theoretical tree-like properties) could be explained more clearly with musical examples or perceptual insights.

The experiments focus on VAEs with hyperbolic vs. Euclidean latent spaces, but the paper does not compare performance with modern large-scale transformer-based music models trained on massive datasets, which are state-of-the-art in generative quality.

It is not shown how the hyperbolic embeddings improve performance on downstream tasks such as style transfer, music recommendation, or generation diversity, limiting the practical impact beyond reconstruction and interpretability.

**Questions:**

See weaknesses

---

> ### Author Response · Authors · 2025-11-15
>
> Thank you for your feedback.
>
> We agree that our core contributions are (i) transferring hyperbolic representations to the music domain to enable natural embeddings which explicitly reflect hierarchical dependencies, and (ii) introducing R-SLERP as a novel interpolation method on Riemannian manifolds that guarantees that the hierarchy levels of interpolated instances stay between those of start- and endpoints. Our experiments demonstrate that hyperbolic representations can be a valuable building block for music processing models with an improved interpretability and quality of internal representations. The potential of R-SLERP ranges beyond the music domain and might also give rise to new applications of Riemannian representations in other domains.
>
> Our experiments focus on an isolated evaluation of our contribution. We acknowledge that there exist other generative models for music that achieve a higher generative quality than our hyperbolic VAE. However, we do not aim at presenting a new SOTA model but intend to provide new conceptual insights. Thus, we compare hyperbolic and Euclidean VAEs while only varying the latent space geometry to isolate its effect and observe that the hyperbolic variant outperforms the Euclidean counterpart. Further, our UMAP and interpolation plots show that the hierarchical structures in music data are explicit in hyperbolic space, encoding musical richness as distance to the origin and pitches as angles.

---

### Official Review · Reviewer_Jhze · 2025-11-01

**Soundness:** 2
**Presentation:** 2
**Contribution:** 2
**Rating:** 4
**Confidence:** 4

**Summary:**

The paper suggested that music’s structure is hierarchical and therefore better modelled in negatively curved latent spaces than in Euclidean ones. Therefore, it defines a VAE whose latent space is a Poincare ball; sampling uses a Wrapped Normal with log/exp maps and parallel transport, and the decoder consumes \log_0(z). The loss is an ELBO with a Monte-Carlo KL. A procedure called “R-SLERP” interpolates by SLERPing unit directions in T_0 and scaling the radius linearly before mapping back with \exp_0. Experiments on POP909, MIDICAPS (symbolic) and MAESTRO (audio) show statistically reconstruction gains over a Euclidean VAE and a latent organization consistent with the circle of fifths.

**Strengths:**

The paper offers a refreshing perspective by attempting to embed musical hierarchies, such as tonality, key relationships, and structural depth within a hyperbolic latent space, where the geometry itself mirrors how music naturally branches and resolves. In essence, the authors use the Poincare ball to represent tonal distance and harmonic tension in a way that feels musically intuitive: the deeper or more nested a tonal relationship, the farther it sits from the origin, much like how modulation in tonal space creates perceptual “depth.” Technically, the model extends the Variational Autoencoder (VAE) into this non-Euclidean space, defining the exponential and logarithmic maps correctly and applying the hyperbolic reparameterization trick to learn latent structures that curve outward.

**Weaknesses:**

a) A key methodological ambiguity arises from the decoding step described in Section 4.2.  After sampling a manifold point z using the Wrapped Normal distribution (Eq. 7), the paper states that “the result of \log_{0}(z) is then decoded in standard fashion to produce \hat{x}”. This means the decoder operates on \log_{0}(z), a vector in the tangent space at the origin T_{0}\mathcal{M}, rather than on the manifold point z itself. Because T_{0}\mathcal{M} is Euclidean, this choice effectively flattens the curvature for the likelihood p_{\theta}(x\mid z), making the decoder independent of the negative curvature that motivates the model.  But, the paper provides no justification for why this operation preserves the claimed benefits of hyperbolic geometry.  The authors should clarify whether decoding from \log_{0}(z) is theoretically equivalent to decoding from z or \log_{\mu}(z), and, if so, under what invariances or assumptions this equivalence holds.  If not, empirical evidence comparing these alternatives is needed to confirm that curvature has a meaningful influence on the generation process, rather than being restricted to the prior–posterior space.  In its current form, the use of \log_{0}(z) risks neutralising the negative-curvature effects in the decoder, undermining the central claim that the proposed Poincare-VAE benefits from a non-Euclidean latent geometry.

---

b) In Section 4.3, the paper introduces R-SLERP as an interpolation method that it claims is “guaranteed to stay within the hierarchy levels of its start and end point,” and that the distance to the origin “changes linearly with t, given that such behaviour holds for v_{\text{int}}(t)”. However, what this  section actually provides is a constructive recipe, that is, it maps the two endpoints z_1, z_2 to the origin’s tangent space with \log_0, performing SLERP on the unit directions, scaling the radius linearly \alpha(t)=(1-t)\|v_1\|+t\|v_2\|, and mapping back using \exp_0, but without a formal derivation or set of sufficient conditions to justify the “guarantee.”

Technically, this construction ensures only that the radius from the origin changes linearly if the tangent-space behaviour meets certain assumptions, because Eq. 4 earlier shows d(0,\exp_0(v))=\|v\|. However, the paper never proves that the resulting path always remains between the two endpoint radii or that it preserves the intended “hierarchy level” for all admissible points in the Poincare ball. In simple terms, the authors are asserting that the curve connecting two musical representations never overshoots or collapses outside their hierarchy range, but they do not actually prove it. They only describe the recipe that should, in theory, make it happen.

This is critical for me because the paper’s entire argument that hyperbolic interpolation captures smooth hierarchical transitions in music depends on this behaviour being guaranteed by geometry, not just observed empirically. To be convincing, the authors need to (a) provide a short lemma showing that d(0,z_{\text{int}}(t))=\alpha(t)\in[d(0,z_1),d(0,z_2)] for all t\in[0,1], or (b) explicitly state that the claim is heuristic. An empirical observation rather than a mathematical fact. Without that, the so-called “hierarchy-preserving” property remains an intuitive but unverified idea, leaving the interpolation step, and thus one of the paper’s main technical contributions, on uncertain theoretical ground.

---
c) In Section 5, it compares only two models: the proposed P-VAE with a Poincare ball latent space and an E-VAE with Euclidean latent space. In the same subsection (“Models”), the authors explicitly state that prior music-VAE work on architectural optimisation and disentanglement (Roberts et al., 2018; Brunner et al., 2018; Li & Mandt, 2018; Yang et al., 2019; Wang et al., 2020b) is “orthogonal” to their goal of comparing hyperbolic vs Euclidean geometry and “may further improve results,” and is therefore not evaluated. They also implement a “standard Euclidean VAE” and note that it does not differ significantly from their E-VAE because the two are mathematically equivalent. As a result, all reported gains are of the form “P-VAE > this one Euclidean baseline,” and the experiments do not isolate curvature as the unique cause of improvement. To make the curvature claim testable, additional Euclidean and non-Euclidean baselines would be required.

That is, all reported improvements hinge on outperforming one basic flat-space model. The omission of stronger Euclidean or alternative-geometry baselines makes it impossible to determine whether the observed gains genuinely arise from negative curvature or merely from unrelated architectural or training differences. For me, this restricted setup cannot substantiate the paper’s central claim that hyperbolic geometry itself improves musical representations. To isolate curvature as the causal factor, the study should expand its baselines to include at least some of below:
(i) advanced Euclidean controls such as β-VAEs, disentangled or hierarchical VAEs, or Euclidean VAEs with regularized Riemannian metrics (e.g., Chen et al., 2022);
(ii) alternative non-Euclidean geometries like spherical or mixed-curvature manifolds (Skopek et al., 2020); and
(iii) explicit ablations varying the curvature constant c while keeping all other parameters fixed.

**Questions:**

- Section 4.3 Header reads “Riemannnian SLERP” (extra ‘n’).

---

> ### Author Response · Authors · 2025-11-15
>
> Thank you for your time and effort invested in the extensive review. We would like to comment on your questions and concerns as follows.
>
> **Decoder acting on Euclidean space.** While the decoder of the VAE acts on the tangent space of the Poincaré ball, which is indeed a Euclidean space, the variational posterior and prior distributions are defined directly on the manifold, and also the KL-divergence term in the loss function is calculated accordingly. This encourages the hierarchical structures of the data to match the geometry of hyperbolic space, and this explicit hierarchical ordering is also preserved when mapped into the tangent space. The effectiveness of this strategy is also demonstrated by previous publications (e.g., Nagano et al., 2019; Mathieu et al., 2019; Skopek et al., 2020).
>
> **Theoretical guarantees of R-SLERP.** R-SLERP guarantees that the geodesic distance to the origin stays between the distances of start- and endpoint to the origin along the entire interpolation path, which follows directly from the distance preserving property of the exponential map. The argument is as follows: We map the start- and endpoint of the interpolation path ($z_1, z_2 \in \mathbb{P}$) to corresponding velocities in the tangent space at the origin of the Poincaré ball ($v_1, v_2 \in T_{0}\mathbb{P}$), and calculate interpolated velocities in this Euclidean space. In particular, we determine the norm of each interpolated velocity as the convex combination $\|v_{\text{int}}(t)\| = \alpha(t) = (1-t) \|v_1\| + t \|v_2\|$, guaranteeing $\min_{t \in [0, 1]}(\|v_{\text{int}}(t)\|) = \min(\|v_1\|, \|v_2\|)$ and $\max_{t \in [0, 1]}(\|v_{\text{int}}(t)\|) = \max(\|v_1\|, \|v_2\|)$. As the the exponential map preserves distances to its development point, i.e. the origin, it is also guaranteed that $\min_{t \in [0, 1]}(d(0, z_{\text{int}}(t))) = \min(d(0, z_1), d(0, z_2))$ and $\max_{t \in [0, 1]}(d(0, z_{\text{int}}(t))) = \max(d(0, z_1), d(0, z_2))$.
>
> **Baselines.** Our experimental setup focuses on the isolated evaluation of our contribution. The $\mathbb{E}$-VAE and $\mathbb{P}$-VAE are both implemented using manifold operations, thus being truly equal except for the latent geometry. Therefore, their comparison indeed allows to measure the effect of switching to a hyperbolic latent space in isolation. The additional comparison of the $\mathbb{E}$-VAE to a standard Euclidean VAE is only intended to verify the correctness of our implementation. As we don’t aim at introducing a new SOTA model but instead focus on an isolated conceptual improvement, we do not consider the inclusion of additional Euclidean or non-Euclidean baselines as valuable.
>
> **References**
>
> Emile Mathieu, Charline Le Lan, Chris J. Maddison, Ryota Tomioka, and Yee Whye Teh. Continuous hierarchical representations with poincaré variational auto-encoders. In Advances in Neural Information Processing Systems 32, pp. 12544–12555, 2019.
>
> Ondrej Skopek, Octavian-Eugen Ganea, and Gary Bécigneul. Mixed-curvature variational autoencoders. In Proceedings of the 8th International Conference on Learning Representations, 2020.
>
> Yoshihiro Nagano, Shoichiro Yamaguchi, Yasuhiro Fujita, and Masanori Koyama. A wrapped normal distribution on hyperbolic space for gradient-based learning. In Proceedings of the 36th International Conference on Machine Learning, pp. 4693–4702, 2019.

---

> ### Comment · Reviewer_Jhze · 2025-11-28
>
> I thank the authors for the rebuttal and the clarifications provided. However, after carefully examining both the response and the manuscript, I find that the rebuttal does not adequately resolve the three central issues raised in my initial review. I outline them below to explain why my concerns remain.
>
> **1. Decoder Geometry: Effectively Euclidean Likelihood**
>
> My original concern was that decoding from log₀(z) places the generative likelihood $p_\theta(x \mid z)$ entirely in the Euclidean tangent space, which risks neutralising the influence of curvature during generation. The paper states explicitly (Sec. 4.2) that *“the result of \log_{0}(z) is then decoded in standard fashion,”* but provides no justification for why decoding from $\log_{0}(z)$ preserves hyperbolic structure or is equivalent to decoding from $z$ or $\log_{\mu}(z)$.
>
> The rebuttal restates that the tangent space is Euclidean and cites prior hyperbolic VAE work, but it does not provide either:
> - a theoretical argument showing the decoder is invariant under this choice, or
> - empirical comparisons of the alternative decoding points, or
> - conditions under which this flattening step still transmits curvature information.
>
> As a result, it remains unclear whether curvature meaningfully influences the generative distribution or is restricted to the prior-posterior space. This ambiguity affects the core claim that the proposed model benefits from non-Euclidean latent geometry.
>
> **2. R-SLERP Guarantee: Unproven Theoretical Claim**
>
> My second concern was that the paper asserts a guarantee that R-SLERP *“stays within the hierarchy levels of its start and end point”* and that the radius changes linearly, but provides only a procedural description (Algorithm 1) rather than a formal derivation or sufficient conditions.
>
> The rebuttal repeats the construction but continues to omit:
> - a lemma demonstrating that $d(0, z_{\text{int}}(t)) = \alpha(t)$,
> - a proof that $\alpha(t) \in [d(0, z_1), d(0, z_2)]$ for all $t$,
> - or an explicit statement acknowledging that this is a heuristic rather than a geometric guarantee.
>
> Given that the interpolation behaviour is presented as a key technical contribution, the lack of a formal justification substantially weakens the theoretical grounding of R-SLERP.
>
> **3. Baseline Design: Curvature Effects Not Isolated**
>
> The paper compares the P-VAE only to a single Euclidean VAE baseline. As previously noted, the authors cite several works on improved VAE architectures (β-VAEs, disentangled models, regularized Riemannian metrics, etc.) but treat them as orthogonal and therefore omit them. This design choice makes it difficult to attribute performance gains to hyperbolic geometry rather than to differences in regularisation, architecture, or optimization dynamics.
>
> The rebuttal reiterates the conceptual focus but does not provide:
> - stronger Euclidean controls,
> - alternative-geometry baselines (e.g., spherical or mixed-curvature models),
> - curvature-sweeping ablations with matched architectures,
> - or any empirical analysis that isolates curvature as the causal factor.
>
> Hence, the central empirical claim that negative curvature itself improves musical representation remains insufficiently supported.
>
> Overall, I appreciate the authors’ explanations and the additional narrative; however, the rebuttal does not adequately address the core theoretical and experimental gaps. The paper presents interesting ideas, and the topic is promising, but the current version does not yet meet the level of rigor required to substantiate its main claims.
>
> My evaluation therefore remains unchanged.

---

> > ### Author Response · Authors · 2025-11-28
> >
> > Thank you for your reply. We would like to comment on your points (1-3) as follows.
> >
> > **(1) Decoding from the (Euclidean) tangent space.** The decoder $p_\theta(x | z)$ utilizes the hyperbolic latent space structure because latent representations $z$ are sampled in hyperbolic space, either from the variational posterior distribution defined by the encoder (for reconstruction), or from the prior distribution (if one would utilize the model for unconditional generation). The hyperbolic latent space reflects the hierarchical relations in music data better than its Euclidean counterpart, and this improved ordering is preserved when mapped to the tangent space at the origin, which is also reflected by the improved reconstruction performances that we observe on all three datasets in our experiments. As stated before, the strategy of mapping representations on manifolds into the tangent space at the origin before decoding was previously shown to be effective in other domains (Nagano et al., 2019; Mathieu et al., 2019; Skopek et al., 2020).
> >
> > **(2) Guarantees of R-SLERP.** Please consider the following argument from our rebuttal which shows that R-SLERP guarantees that the distances to the origin of interpolated points stay between those of start- and endpoint: The norm of each interpolated velocity in the tangent space at the origin of the manifold $v_{\text{int}} \in T_0M$ is determined as the convex combination $\|v_{\text{int}}(t)\| = \alpha(t) = (1-t) \|v_1\| + t \|v_2\|$, guaranteeing $\min_{t \in [0, 1]}(\|v_{\text{int}}(t)\|) = \min(\|v_1\|, \|v_2\|)$ and $\max_{t \in [0, 1]}(\|v_{\text{int}}(t)\|) = \max(\|v_1\|, \|v_2\|)$. As the the exponential map preserves distances to its development point, i.e. the origin, it follows for the interpolated points on the manifold $z_{\text{int}} \in M$ that $\min_{t \in [0, 1]}(d(0, z_{\text{int}}(t))) = \min(d(0, z_1), d(0, z_2))$ and $\max_{t \in [0, 1]}(d(0, z_{\text{int}}(t))) = \max(d(0, z_1), d(0, z_2))$.
> >
> > **(3) Isolation of the effect of latent space curvature.** As stated in the rebuttal, the $\mathbb{E}$-VAE and $\mathbb{P}$-VAE are equal except for their latent space geometry. Thus, our experiments directly compare the use of hyperbolic geometry (negative curvature) vs. Euclidean geometry (curvature of 0) in isolation. Further, we would like to highlight that all VAEs are implemented as $\beta$-VAEs, whereby $\beta$ was tuned for all models in a grid search.
> >
> > **References**
> >
> > Emile Mathieu, Charline Le Lan, Chris J. Maddison, Ryota Tomioka, and Yee Whye Teh. Continuous hierarchical representations with poincaré variational auto-encoders. In Advances in Neural Information Processing Systems 32, pp. 12544–12555, 2019.
> >
> > Ondrej Skopek, Octavian-Eugen Ganea, and Gary Bécigneul. Mixed-curvature variational autoencoders. In Proceedings of the 8th International Conference on Learning Representations, 2020.
> >
> > Yoshihiro Nagano, Shoichiro Yamaguchi, Yasuhiro Fujita, and Masanori Koyama. A wrapped normal distribution on hyperbolic space for gradient-based learning. In Proceedings of the 36th International Conference on Machine Learning, pp. 4693–4702, 2019.

---

### Official Review · Reviewer_ZXJH · 2025-11-07

**Soundness:** 2
**Presentation:** 3
**Contribution:** 1
**Rating:** 2
**Confidence:** 4

**Summary:**

This paper proposes learning hyperbolic representation spaces for music using variational autoencoders with a Poincaré ball latent space. The authors motivate this by claiming that musical structures are hierarchical and thus, better suited to hyperbolic geometry than to Euclidean space. They also introduce a Riemannian generalization of spherical linear interpolation (R-SLERP) to generate interpolations that vary linearly in hierarchy level along the path. Experiments on POP909, MIDICAPS, and MAESTRO show slightly improved reconstruction metrics over a Euclidean baseline and visually coherent interpolations.

**Strengths:**

- The paper provides a coherent geometric narrative linking hierarchical structures to negative curvature, mapping this intuition to music in an interpretable manner.

- Writing and visuals are clear. The figures and geometric explanations make the paper approachable for readers less familiar with hyperbolic geometry.

- R-SLERP addresses a real issue with geodesic interpolation. The observation that geodesics curve toward the origin (causing “over-simplified” midpoints) is valid. The proposed interpolation is a reasonable workaround. The qualitative results provided, though limited, support this claim.

- The experimental setup covers both symbolic and raw waveform music datasets, and the reported reconstruction improvements are small but statistically significant.

**Weaknesses:**

- Limited technical contribution. The Poincaré VAE and wrapped-normal sampling closely follow prior work (Nagano et al., 2019). R-SLERP is essentially a composition of log/exp maps and SLERP, with linear radial scaling. While it helps to avoid interpolation paths curving toward the origin, it does not constitute a clear theoretical advance or provide formal guarantees.

- Speculative motivation. The claim that music is hierarchical rests on the combinatorial growth of note sequences, which applies to any sequential data. The dependency that “each note constrains the next” is not unique to music. The connection between musical structure and tree-like geometry, or negative curvature, is not formally or empirically substantiated.

- Marginal empirical gains. The claim that hyperbolic representations are superior is not convincingly supported. Reconstruction improvements (Table 1) are small, yet the authors describe them as “impressively” demonstrating the superiority of hyperbolic latent spaces. Furthermore, the hyperbolic VAE increases runtime by 7–40%, and the latent dimensionality is identical to Euclidean baselines (128d for POP909, MAESTRO and 512d for MIDICAPS), undermining the expected efficiency benefits of hyperbolic representations.

- Conceptual tension around interpolation. The authors argue that hyperbolic geometry is a natural fit for musical hierarchy, yet their principal interpolation justification is that hyperbolic geodesics are a poor choice (they curve inward, producing sparse midpoints). The authors must explain why choose hyperbolic geometry in the first place.

- Insufficient experimental validation. R-SLERP is the main technical novelty, yet there is only 1 qualitative experiment presented. In this experiment, the results seem to agree with the hypothesis that hyperbolic geodesic interpolation lead to sparse midpoints. However, no comparison was made to Euclidean interpolation (Euclidean VAE) and there's no quantitative results to conclude that R-SLERP produces better or more meaningful interpolations.

- Interpretability claims: The UMAP plot in Fig. 4 is interesting but there's no comparison against a Euclidean baseline to assess whether the angular separation and arrangement of keys is due to hyperbolic geometry. The same follows for the claim that radius encodes richness. There is a single example of this in Fig. 5c. Consequently, it's unclear whether the proposed approach is also better in terms of  interpretability. Quantitative validation (e.g., classifier for key from angular coordinate, correlation between radius and richness metrics) and comparison against Euclidean baselines is needed.

**Questions:**

- Can the authors provide a clearer justification for why hyperbolic geometry is the appropriate choice for modeling musical hierarchies, especially given that geodesics in hyperbolic space behave poorly for interpolation?

- Can the relationship between radial distance in the latent space and objective measures of musical richness (e.g., chord size, number of unique notes) be quantified?

- Did the authors experiment with lower-dimensional latent spaces to test whether hyperbolic representations preserve reconstruction quality better than Euclidean representations at smaller dimensionalities?

- Could the authors provide a UMAP (or other dimensionality reduction) visualization of the Euclidean VAE latent space for comparison with the hyperbolic latent space?

---

> ### Author Response · Authors · 2025-11-15
>
> Thank you for the extensive review.
>
> The core motivation for representing music in hyperbolic space is the observation that there exist hierarchical relations between different sequences of music. However, this hierarchy does not emerge solely from the trivial combinatorial growth that every sequence exhibits as it grows in length, but additionally from the varying numbers of notes that can be played simultaneously in each time step. That is, one can view a music sequence as a sequence of sets of varying sizes, which also leads to hierarchical relations between music sequences of the same length (in theory, even for a single time step). If one exemplarily considers a short melodic pattern, one could play this pattern with varying accompaniments, and the original pattern would always be a subset of the enriched versions.
>
> The fit between hierarchical data and hyperbolic geometry arises from the alignment between the growth of tree-like structures and hyperbolic space. Therefore, one expects instances at high levels of the hierarchy (instances with few notes for music) to be located close to the origin, and instances occupying low hierarchy levels more distant. Such an organization of hierarchical structures in hyperbolic space was for example also demonstrated by Mathieu et al. (2019) or Kiela and Nickel (2017). As geodesics bend towards the origin in hyperbolic space, it is not contradictory but expected that one traverses through higher hierarchy levels as one interpolates along geodesics. However, while interpolating along geodesics is the natural generalization of linear interpolations to Riemannian manifolds, the resulting interpolation paths are impractical for music applications. Therefore, we propose R-SLERP as a more useful alternative.
>
> R-SLERP guarantees that the geodesic distance to the origin stays between the distances of start- and endpoint to the origin along the entire interpolation path, which follows directly from the distance preserving property of the exponential map. The argument is as follows: We map the start- and endpoint of the interpolation path ($z_1, z_2 \in \mathbb{P}$) to corresponding velocities in the tangent space at the origin of the Poincaré ball ($v_1, v_2 \in T_{0}\mathbb{P}$), and calculate interpolated velocities in this Euclidean space. In particular, we determine the norm of each interpolated velocity as the convex combination $\|v_{\text{int}}(t)\| = \alpha(t) = (1-t) \|v_1\| + t \|v_2\|$, guaranteeing $\min_{t \in [0, 1]}(\|v_{\text{int}}(t)\|) = \min(\|v_1\|, \|v_2\|)$ and $\max_{t \in [0, 1]}(\|v_{\text{int}}(t)\|) = \max(\|v_1\|, \|v_2\|)$. As the the exponential map preserves distances to its development point, i.e. the origin, it is also guaranteed that $\min_{t \in [0, 1]}(d(0, z_{\text{int}}(t))) = \min(d(0, z_1), d(0, z_2))$ and $\max_{t \in [0, 1]}(d(0, z_{\text{int}}(t))) = \max(d(0, z_1), d(0, z_2))$.
>
> **References**
>
> Emile Mathieu, Charline Le Lan, Chris J. Maddison, Ryota Tomioka, and Yee Whye Teh. Continuous hierarchical representations with poincaré variational auto-encoders. In Advances in Neural Information Processing Systems 32, pp. 12544–12555, 2019.
>
> Maximilian Nickel and Douwe Kiela. Poincaré embeddings for learning hierarchical representations. In Advances in Neural Information Processing Systems 30, pp. 6338–6347, 2017.

---

### Official Review · Reviewer_pQFM · 2025-11-12

**Soundness:** 4
**Presentation:** 3
**Contribution:** 2
**Rating:** 4
**Confidence:** 3

**Summary:**

This paper suggests that there is a fundamental incongruity with the nature of music and the practice of embedding music in a Euclidean space.  The authors suggest that a Riemannian space is more appropriate, and then show how to adopt the widely-used VAE technique of embedding data into R^n to study this phenomenon using Poincare balls.  They show that they do better on reconstruction scores than traditionally-trained VAEs, and that they are able to meaningfully interpolate in the induced space.

**Strengths:**

This paper applies a SOTA realization from the literature - that tree-like structures are better embedded in Riemanniam than Euclidean spaces to the oft-neglected topic of music.  The UMAP projection very nicely demonstrates that there is real mathematical backing for the “circle of 5ths” hypothesized in music theory.  The SLERP VAE method is novel and could be impactful in other areas as well.

**Weaknesses:**

This paper is not particularly high-impact in terms of improving content in the application domain - there are much, much better symbolic transformers than the MusicVAE.  While I recognize that science needs basic experiments, I’m not convinced by these - the interpolation examples give didn’t look particularly meaningful (I honestly didn’t know what I was supposed to be looking for).  There could have been much more statistical rigor.

**Questions:**

1. Could this method be applied to more than just short snippets of music (which was an inherent limitation of VAE’s) in the past?
2. Could you explain what I should be looking for in the interpolation figures?
3. Are reconstruction losses between the riemannian and euclidean domain directly comparable like you seem to claim?
4. Have you been able to use VAE output to do any musical property classification?
5. What would you say could be taken from your work and applied to music \textbf{generation} (not theory), if anything?

---

> ### Author Response · Authors · 2025-11-15
>
> Thank you for your feedback and questions.
>
> We acknowledge that there exist other generative models for music that achieve a higher generative quality than our hyperbolic VAE. However, we do not aim at presenting a new SOTA model but intend to provide new conceptual insights:  To the best of our knowledge, our study is the first that embeds music into hyperbolic space to explicitly capture hierarchical relations between pieces of music. In the experiments, we compare hyperbolic and Euclidean VAEs while only varying the latent space geometry to isolate its effect and observe that the hyperbolic variant outperforms the Euclidean counterpart.
>
> **Applications to longer sequences/music generation tasks.** There are different possibilities to apply our approach to music (generation) tasks involving longer sequences. For example, the music VAE presented by Roberts et al. (2018) can generate pieces with a length of 16-bars (around 30 seconds), using a hierarchical decoder architecture. The hyperbolic latent space can be readily integrated into this architecture, as the change of the latent geometry is independent of the encoder and decoder architecture. Another strategy would be to utilize sequences of hyperbolic embeddings as inputs for subsequent model components. Following this strategy, Mittal et al. (2021) use a latent diffusion model that acts on a sequence of VAE-based (Euclidean) music embeddings. A similar strategy could also be chosen to utilize hyperbolic embeddings for other downstream tasks beyond music generation.
>
> **Interpolations.** The interpolation plots validate the expected latent space structure and demonstrate the use of R-SLERP. Points that are close to the origin of the Poincaré ball correspond to high hierarchy levels and thus represent instances with few notes (and points that are far from the origin vice versa), which is visualized via the interpolation from the origin towards an exemplary instance. Further, interpolations along geodesics bend towards the origin in hyperbolic space and thus yield increasingly sparse instances towards the midpoint of the interpolation path, which is impractical for many applications. R-SLERP, in contrast, ensures that the distance to the origin stays between start- and endpoint along the interpolation path.
>
> **Comparability of reconstruction losses.** The reconstruction loss is calculated in input space for all models and thus comparable across latent space geometries.
>
> **References**
>
> Gautam Mittal, Jesse H. Engel, Curtis Hawthorne, and Ian Simon. Symbolic music generation with diffusion models. In Proceedings of the 22nd International Society for Music Information Retrieval Conference, pages 468–475, 2021.
>
> Adam Roberts, Jesse H. Engel, Colin Raffel, Curtis Hawthorne, and Douglas Eck. A hierarchical latent vector model for learning long-term structure in music. In Proceedings of the 35th International Conference on Machine Learning, pp. 4361–4370, 2018.

---

### Note · Authors · 2026-01-15

I have read and agree with the venue's withdrawal policy on behalf of myself and my co-authors.